# Intestinal Microbiota of Commercial Laying Hens in a Cage-Free System Fed with Probiotics

**DOI:** 10.3390/ani15233388

**Published:** 2025-11-24

**Authors:** Weslane Justina da Silva, Josilene Silva Sousa, Nadielli Pereira Bonifácio, Murilo Sousa Carrijo, Cíntia Minafra, Hindenburg Cruvinel Guimarães da Costa, Fabiana Ramos dos Santos, Cassia Cristina Fernandes, Adriano Carvalho Costa, Sérgio Turra Sobrane Filho, Fabiano Guimarães Silva, Cibele Silva Minafra

**Affiliations:** 1Program Postgraduate in Biotechnology and Biodiversity of Goiás—PGBB/UFG, Federal University of Goiás, Samambaia Campus, Goiânia 74001-240, Brazil; weslanejds@gmail.com; 2Goiano Federal Institute of Education, Science and Technology (Federal Institute Goiano–IF Goiano), Rio Verde 75901-970, Brazil; josilene.silva@estudante.ifgoiano.edu.br (J.S.S.); nadielli@yahoo.com.br (N.P.B.); diretoria@laboratoriohormonal.com.br (H.C.G.d.C.); fabiana.santos@ifgoiano.edu.br (F.R.d.S.); cassiacefetrv@gmail.com (C.C.F.); adriano.costa@ifgoiano.edu.br (A.C.C.); fabiano.silva@ifgoiano.edu.br (F.G.S.); 3Faculty of Veterinary Medicine, Fontes do Saber Farm—University Campus, University of Rio Verde, University District, Rio Verde 75901-970, Brazil; carrijoms@hotmail.com; 4School of Veterinary Medicine, Goiânia Highway, Federal University of Goiás—UFG, km 8 s/n Campus—Samambaia, Goiânia 74001-970, Brazil; cintia.minafra@ufg.br; 5Institute of Agricultural Sciences, Federal University of Jataí, BR-364 Highway, Francisco Antônio District, Jataí 75801-615, Brazil; sergioturra@ylive.com.br

**Keywords:** bacteria, *Helicobacter brantae*, Isa Brown, laying hens, *Lactobacillus* spp.

## Abstract

The market niche for cage-free chicken production is favorable and expanding, as consumers are becoming more demanding regarding products originating from systems that ensure animal welfare. In this context, the cage-free production system has been adopted, allowing hens raised without cages to express natural behaviors. However, several factors can impact alternative production systems, and the appropriate inclusion of probiotics can improve efficiency, reduce morbidity, decrease mortality and environmental pollution, and enhance food safety. Probiotics are expected to have beneficial effects like those of antibiotics used in conventional poultry production systems. In addition, this practice encourages and strengthens family farming, which presents several advantages: the use of poor and undervalued land for implementing the activity, low investment in facilities and equipment, improvement of the producer’s and family’s diet through increased protein intake, retention of the producer on the property, and provision of supplementary income to the rural enterprise budget.

## 1. Introduction

Laying hens raised cage-free are happier because they can express natural species-specific behaviors, such as dust bathing, perching, scratching, flapping, and wing stretching, thus ensuring the hens’ welfare and making production more sustainable [1].

In addition to the poultry housing system, nutrition also plays a pivotal role, as the type and origin of feed ingredients directly reflect consumers’ growing concerns regarding sustainability and ethical production practices [2,3]. In other words, animal welfare and feeding practices are key factors in attracting consumers of animal-derived products, as animals that can express their natural behaviors during production tend to increase market acceptance of their products [4,5].

The feeding of laying hens involves providing nutritionally balanced diets and incorporating additives that help prevent disease occurrence; these additives act as promoters of health and productivity in the animals [6,7]. Alternative additives to antibiotics, in addition to promoting health and efficient nutrient utilization, also reflect responsible production practices, environmental sustainability, and economic viability, thereby driving the poultry industry [8,9,10].

In this context, an interesting option is the inclusion of probiotics in poultry diets, as they consist of various beneficial bacterial species, fungi, or yeasts that not only promote animal growth but also inhibit pathogenic microorganisms, such as *Salmonella* Typhimurium, *Staphylococcus aureus*, *Escherichia coli*, and *Clostridium perfringens* [11,12]. The inclusion of probiotics in laying hen diets enhances nutrient absorption, positively impacts intestinal health, and improves feed efficiency and overall performance through microbiota modulation [4,13].

The intestinal microbiota of laying hens performs essential functions, including protection against pathogens, nutrient synthesis, and modulation of the immune system, directly influencing gut health [14]. The balance of microbial populations in the avian gut can be affected by factors such as diet, heat stress, and inadequate management, leading to dysbiosis and negatively impacting bird health and productivity [15].

Given this context, the aim of the present study was to evaluate the effect of adding probiotics on the composition and diversity of the gut microbiota of Isa Brown laying hens reared in a cage-free system.

## 2. Materials and Methods

The experiment was conducted at the Instituto Federal Goiano, Rio Verde Campus—GO, in the Poultry Sector and in the Animal Nutrition and Biochemistry and Animal Metabolism Laboratories, following approval by the Research and Graduate Studies Directorate of the Instituto Federal Goiano and the Ethics Committee on the Use of Animals (CEUA) under protocol CEUA nº 5032310724.

A total of 450 Isa Brown laying hens aged 19 weeks old were maintained in a cage-free system with free access to nests, and the barn floor was covered with wood shavings. The experiment lasted 84 days and was divided into three 28-day cycles, with ad libitum access to water and the experimental diets.

An 18 h lighting program was adopted, combining natural and artificial light sources: natural light from sunrise to sunset, complemented by LED lamps with cool white light and a luminous flux of 1800 lumens, which remained on from 6 a.m. to midnight, controlled by a timer that automatically turned the lamps on and off according to the programmed schedule. Temperature and humidity were measured using a thermohygrometer, with averages recorded in the morning and afternoon. Environmental conditions were monitored throughout the trial, with maximum temperatures ranging from 28.62 ± 1.42 °C in the morning to 28.84 ± 1.05 °C in the afternoon. Relative humidity reached high maximum values, with 90.19 ± 7.17% in the morning and 87.38 ± 6.15% in the afternoon. The minimum temperature recorded during the experimental period was 21.93 ± 1.44 °C in the morning and 22.33 ± 2.26 °C in the afternoon, while the minimum relative humidity was 64.42 ± 7.61% in the morning and 60.95 ± 7.23% in the afternoon. These data indicate that the environmental conditions in the barn were within acceptable limits for the rearing of Isa Brown laying hens in a cage-free system [16,17].

### 2.1. Housing of the Hens

The facilities and materials used during rearing were in accordance with the Certified Humane Brazil Institute, representative of Humane Farm Animal Care (HFAC) [16]. The laying hens were housed in a masonry shed with a zinc tile roof, wire mesh sides with curtains, and internal divisions made of wire mesh. Each pen measured 2 m in length, 1 m in width, and 1.80 m in height. The floor was made of concrete and covered with wood shavings as litter. Each pen was equipped with a nipple-type drinker, a pendular feeder (although the feeders had a capacity of up to 5 kg, only 2 kg of feed were provided daily, served twice a day (1 kg at 8 a.m. and 1 kg at 4 p.m.)), and two nests. The nests were adapted from plastic baskets lined with pieces of synthetic grass. Each nest measured 31 cm in length, 34 cm in width, and 31 cm in height, arranged in two stacked levels, as shown in the following Figure 1.

### 2.2. Diet

The experiment was conducted in a completely randomized design (CRD), consisting of five treatments with six replicates, each containing 15 laying hens per pen. The treatments were as follows for isonutritive diets: basal ration with corn and soybean meal; basal ration with the addition of a single probiotic strain, Colostrum^®^ BS (Powder formulation) composed of *Bacillus subtilis* (LOFU 160) at 500 g/t; basal ration supplemented with a single strain in the proportion of 1000 g/t (*Bacillus subtilis* (LOFU 160)); basal ration with the inclusion of a probiotic mixture, Colostrum^®^ BIO 21 MIX (Powder formulation) consisting of *Enterococcus faecium* (LOFU 84, C LOFU 124, LOFU 140, LOFU 145, LOFU 146, LOFU 158), *Lactobacillus acidophillus* (LOFU 63), *Lactobacillus delbrueckii* (LOFU 65) *Lactobacillus plantarum* (LOFU 79, LOFU 83, LOFU 147) *Lactobacillus reuteri* (LOFU 18, LOFU 21, LOFU 22, LOFU 35, LOFU 48) *Lactobacillus salivarius* (LOFU 44) *Pediococcus acidilactici* (LOFU 57, LOFU 58, LOFU 82) *Bacillus subtilis* (LOFU 160) at 200 g/t; and basal ration with inclusion the probiotic blend at 400 g/t (Colostrum^®^ BIO 21 MIX).

The diet was formulated according to the nutritional recommendations of the Brazilian Table, 2017 [18], as presented in Table 1 below:

### 2.3. Intestinal Microbiota Analysis

One hen per replicate was sacrificed by cervical dislocation, followed by jugular incision. Intestinal contents from the small intestine, including the duodenum, jejunum, and ileum, were collected, placed in sterile labeled tubes, and stored at −80 °C. Samples were subsequently lyophilized, and DNA was extracted to profile the intestinal microbiota according to Christoff et al. (2017) [19].

Cell lysis and DNA extraction were performed using the magnetic bead technique 16S rRNA (NeoRef, Neoprospecta Microbiome Technologies, Florianópolis, Brazil). Bacterial communities were characterized by high-throughput sequencing of the V3/V4 regions of the 16S gene employing a MiSeq Sequencing System (Illumina Inc., San Diego, CA, USA). Sequencing was conducted using the 300-cycle V2 kit, single-ended mode, without library normalization.

The resulting DNA sequences were processed using a bioinformatics pipeline developed by Neoprospecta Microbiome Technologies (Florianópolis, Brazil) that adopted an assay applying the NGS technique for bacteria—16S amplicon sequencing—30,000 reads, which integrates quality filtering, chimera removal, and taxonomic assignment steps.

Species-level identification was achieved by comparing the obtained sequences against the JSpecies Web Server database [20] and other reference sequences of relevant microbial taxa. Results were summarized descriptively using the Neobiome platform.

### 2.4. Statistical Analysis

To assess microbial diversity and estimate the similarity of community structure among different diets, microbial identification and quantification were performed based on relative abundance data (%) generated by Neoprospecta/Neobiome, considering phylum, class, order, family, genus, and species. Graphs were constructed to visually present the results.

For identified species, an analysis of variance (ANOVA) was conducted to evaluate statistical differences in abundance among treatments. The means with statistical differences in ANOVA were previously transformed by the Arc Sine (Angular) method because the proportion data were restricted to the range 0% to 100%. Immediately after, Fisher’s Least Significant Difference (LSD) test was applied using the SISVAR^®^ 5.6 statistical software [21].

Fisher’s LSD method calculates the simultaneous confidence level for all confidence intervals by considering the individual error rate and the number of comparisons. This simultaneous confidence level represents the probability that all confidence intervals contain the true difference [22].

## 3. Results

### Intestinal Microbiota at the Phylum Level

The small intestine (including the duodenum, jejunum, and ileum) microbiota of laying hens was characterized by 16S rRNA gene sequencing, targeting the 16S ribosomal RNA genes. The most abundant microbial phylum, class, order, family, and genus levels in the intestinal lumen are presented in Figure 2.

The results for the phyla detected in the intestinal content of laying hens revealed the presence of three phyla: Actinobacteria, Bacillota, and Proteobacteria. In the treatment with basal ration, without probiotic addition, Bacillota was the most abundant phylum, followed by Proteobacteria, while Actinobacteria was present at the lowest proportion.

As shown in Figure 2 panel A, the phylum identified across treatments was represented as 7.25% Actinobacteria in the basal ration and 1.83% in the diet with the inclusion of 200 g/t of the probiotic blend, while being absent in the other treatments.

The phylum Proteobacteria accounted for 32.43% in the basal diet, but its relative abundance decreased sharply to 1.29% in the diet inclusion with 200 g/t of the probiotic blend, and it was virtually absent in the treatments containing the single-strain probiotic and in the diet with 400 g/t of the probiotic blend. The proportion of Bacillota was 60.3% in the treatment with basal ration and 96.9% in the treatment with 200 g/t of the blend.

In contrast, the single-strain probiotic stimulated an increase in bacteria belonging to the phylum Proteobacteria, reaching 89% in the diet with 500 g/t of *B. subtilis* (CCBC 1001) and 88% with 1000 g/t. The probiotic blend, however, reduced the abundance of this phylum to 1% in the diet with 200 g/t and to 5% in the diet containing 400 g/t of the blend.

The identified classes are presented in Figure 2, panel B. For the class Bacilli, a substantial decrease was observed when the single-strain probiotic was added to the diet, declining from 39.87% in the treatment with basal ration to only 4.97% in the diets containing 500 g/t and 1000 g/t of *Bacillus subtilis* (CCBC 1001). In contrast, significant increases were observed with the probiotic blend, reaching 77.33% in the diet with 200 g/t and 91.23% in the diet with 400 g/t of the blend.

The class Clostridia was also identified, showing a marked decrease in the diet containing 400 g/t of the probiotic blend (2.14%), while the treatment with basal ration and the diet with 200 g/t of the blend presented similar values (19.05%). Other probiotic treatments showed low proportions, with 5.77% in the diet with 500 g/t of the single-strain probiotic and 7.88% in the diet with 1000 g/t.

The class Epsilonproteobacteria increased considerably in the single-strain *B. subtilis* treatments, reaching 88.54% (500 g/t) and 88.24% (1000 g/t), but decreased drastically with the probiotic blend, reaching 1.16% (200 g/t) and 5.24% (400 g/t), compared with 32.42% in the treatment with basal ration.

The order-level composition of the microbial communities is presented proportionally in Figure 2, panel C. Approximately six microbial orders were detected. Bifidobacteriales were present at low proportions in the diet containing 200 g/t of the probiotic blend (1.44%) and in the treatment with basal ration (7.14%), and they were absent in the other treatments.

The order Clostridiales showed reduced proportions with the addition of the single-strain probiotic (500 g/t and 1000 g/t) and with 400 g/t of the probiotic blend, reaching 5.77%, 7.8%, and 2.4%, respectively. The diet with 200 g/t of the probiotic blend presented a proportion like the treatment with the basal ration (19%).

Campylobacterales was highly represented in the single-strain probiotic treatments, with both inclusion levels (500 g/t and 1000 g/t) showing relative abundances above 88%. In contrast, the probiotic blend markedly reduced the abundance of this order, showing only 1.44% in the 200 g/t treatment and 5.25% in the 400 g/t treatment, compared with 32% in the treatment with basal ration. For Erysipelotrichales, the probiotic blend treatments exhibited low relative abundance, with a maximum of approximately 1.1%, compared with 2% in the treatment with basal ration and 3% in the 1000 g/t single-strain treatment, whereas this bacterial order was not detected in the 500 g/t single-strain treatment. The order Lactobacillales showed distinct proportions across diets. Low abundances were observed in the single-strain treatments (4.70% for 500 g/t and 1.25% for 1000 g/t), lower than the basal diet (39.68%). In contrast, the probiotic blend resulted in higher proportions, reaching 76.91% in the 200 g/t treatment and 91.17% in the 400 g/t treatment.

The microbial families identified are illustrated in Figure 2, panel D. Regarding microbial families, Bifidobacteriacea represented 7.1% in the basal diet, was absent in the single-strain probiotic treatments (500 and 1000 g/t) and in the 400 g/t probiotic blend treatment, and reached 1.4% in the 200 g/t probiotic blend treatment. Clostridiaceae represented approximately 1.86% in the basal diet and 3.17% in the single-strain 1000 g/t and 200 g/t blend treatments only.

Erysipelotrichaceae occurred at low levels, representing 1.64% in the treatment with basal ration, being absent in the single-strain probiotic treatments, and showing 0.5% and 1.1% in the probiotic blend treatments (200 and 400 g/t, respectively). Helicobacteraceae was highly represented in the single-strain treatments, reaching 88.54% and 88.07% in the 500 g/t and 1000 g/t treatments, respectively. In contrast, the probiotic blend resulted in very low proportions, with 1% in the 200 g/t treatment and 3% in the 400 g/t treatment, compared with 33% in the treatment with the basal ration.

Lactobacillaceae had an average proportion of 39.31% in the basal diet. When included with the single-strain probiotic, lower values were observed (4.76% for 500 g/t and 1.16% for 1000 g/t). The probiotic blend, however, increased its abundance to 77% and 91%, making it one of the most abundant families detected.

Peptostreptococcaceae was also quantified, representing 16.94% in the treatment with basal ration, like the 15.87% observed in the diet with 200 g/t of the probiotic blend. In contrast, the 500 g/t and 1000 g/t single-strain treatments and the 400 g/t blend treatment had proportions of 5.49%, 7.49%, and 1.95%, respectively.

The genus-level results, shown in Figure 2, panel E, revealed a marked increase in *Helicobacter* in the single-strain probiotic treatments, reaching 88.54% and 88.07% in the 500 and 1000 g/t treatments, respectively. In contrast, lower proportions were observed in the probiotic blend treatments, with 1.16% and 2.42% in the 200 and 400 g/t treatments, respectively, compared with 32.40% in the treatment with the basal ration.

On the other hand, Lactobacillus was more abundant in the probiotic blend treatments, with 76.69% in the diet containing 200 g/t and 90.95% in the diet with 400 g/t, whereas lower proportions were observed in the single-strain treatments (4.76% and 1.16%), compared with 39.31% in the treatment with basal ration.

*Romboutsia* represented 17.00% of the microbiota in the basal diet and increased to 26.00% in the diet containing 200 g/t of the probiotic blend. In contrast, lower proportions were observed in the single-strain probiotic treatments, with 5.00% and 8.00% in the 500 and 1000 g/t *Bacillus subtilis* diets, respectively, while the genus was not detected in the 400 g/t probiotic blend treatment. *Turicibacter* showed low relative abundance, representing 0.10% in the single-strain probiotic treatment (500 g/t), 2.00% in the treatment with basal ration, and 3.00% in the 1000 g/t single-strain treatment. *Clostridium* accounted for 3.00% in both the basal diet and the diet containing 200 g/t of the probiotic blend, being absent in the other treatments. Similarly, *Aeriscardovia* was detected only in the basal diet (7.00%) and in the 200 g/t probiotic blend treatment (2.00%).

The bacterial species identified in the gut of laying hens are shown in Figure 3, which presents a heatmap of the relative abundance (expressed as *z-scores* per species) and hierarchical clustering of samples. Each column represents an individual sample within each treatment, and the clustering pattern reflects similarities in microbial composition among treatments.

Bacterial species were identified and quantified as relative percentages. In the treatment with basal ration on corn and soybean meal, without probiotic inclusion, the most abundant species were *Helicobacter brantae* and *Romboutsia timonensis*, followed by *Lactobacillus kitasatonis* (17%). Other *Lactobacillus* species were present at lower proportions: 1% for *Lactobacillus reuteri, Lactobacillus vaginalis*, and *Lactobacillus alvi*; 2% for *Lactobacillus aviaries* and *Lactobacillus delbrueckii*; and 4% for *Lactobacillus johnsonii*. Overall, the genus Lactobacillus represented 29% of the bacterial population.

In the diet supplemented with 500 g/t of the single-strain probiotic, *H. brantae* dominated at 87%, *R. timonensis* accounted for 5%, and *Helicobacter kayseriensis* and *Lactobacillus crispatus* were each present at 2%.

Similarly, in the 1000 g/t single-strain treatment, *H. brantae* represented 86%, *R. timonensis* increased slightly to 8%, *H. kayseriensis* remained at 2%, and *Turicibacter sanguinis* was detected at 2%.

In the diet containing 200 g/t of the probiotic blend, *L. crispatus* represented 37% and *L. aviaries* 21%; combined with other *Lactobacillus* species, the genus accounted for 78%, indicating its predominance in this treatment. *R. timonensis* was present at 16%, *Clostridium disporicum* at 3%, and *H. brantae*, *T. sanguinis*, and *Aeriscardovia aeriphila* each at 1%.

In the diet with 400 g/t of the probiotic blend, *L. kitasatonis* accounted for 68%, and *L. crispatus* accounted for 16%, with *Lactobacillus* species overall representing 91% of the bacterial population. Other species were present at lower proportions: *T. sanguinis* 1%, *H. brantae* 2%, *R. timonensis* 2%, and *Campylobacter jejuni* 3%.

In addition to quantifying the species, statistical analyses were performed to compare the average differences between treatments for each species, with the aim of evaluating how the inclusion of probiotics in the chickens’ diet influenced bacterial abundance. This approach allowed investigation of the effects of probiotics and the determination of which formulation—single-strain or blend—is most suitable for beneficial modulation of the intestinal bacterial population in laying hens.

Accordingly, Table 2 presents the mean relative abundances of bacterial species identified in the small intestine of laying hens fed the experimental diets.

The results presented in Table 2 showed statistically significant differences in the relative abundances of microorganisms among the treatments.

*Helicobacter brantae* was highly abundant in the single-strain probiotic treatments, with 86.04% in the 500 g/t treatment and 85.5% in the 1000 g/t treatment, while its abundance decreased drastically in the probiotic blend treatments, with 1.16% in the 200 g/t diet and 2.42% in the 400 g/t diet compared to the treatment with basal ration.

The second most abundant species was *Lactobacillus kitasatonis*, reaching 67.58% in the 400 g/t probiotic blend treatment. The lowest abundance was observed in the 1000 g/t single-strain treatment, with only 0.043%, compared to 14.15% in the treatment with basal ration.

*Lactobacillus crispatus* showed a significant increase in the 200 g/t probiotic blend treatment (36.46%) compared with the treatment with basal ration, while lower abundances were observed in the basal diet and 500 g/t single-strain treatment (approximately 0.35% each). Another species with a notable increase in the 200 g/t probiotic blend treatment was *Lactobacillus aviarius* (36.46%).

In general, other bacteria also showed significant changes compared to the treatment with the basal ration. The 500 g/t single-strain treatment increased the abundance of *Helicobacter kayseriensis*, while the 1000 g/t single-strain treatment increased both *Turicibacter sanguinis* and *H. kayseriensis*.

The 200 g/t probiotic blend treatment increased the abundances of *Aeriscardovia aeriphila, Campylobacter jejuni*, *Lactobacillus acidophilus*, *L. delbrueckii*, *L. ingluviei*, *L. johnsonii*, *L. pontis*, *L. salivarius*, and *Romboutsia timonensis*. The 400 g/t blend treatment stimulated increases in *C. jejuni*, *L. acidophilus*, *L. reuteri*, and *L. vaginalis*.

Significant reductions were also observed for certain bacteria depending on probiotic supplementation. Both the single-strain treatments and the 400 g/t blend treatment decreased the abundances of *L. alvi*, *L. delbrueckii*, *L. ingluviei*, *L. johnsonii*, *L. reuteri*, *L. salivarius*, *T. sanguinis*, and *R. timonensis*.

Significant decreases were also observed for *L. acidophilus* in the single-strain treatments. *L. pontis* and *L. vaginalis* were lower in the single-strain and 200 g/t blend treatments. Similarly, *Romboutsia* sp. showed reduced abundances in the 500 g/t single-strain and 400 g/t blend treatments.

Table 3 presents the correlation coefficients (factor loadings) between microbial variables and the principal components (PCs) obtained from principal component analysis (PCA). Component 1, explaining 54.78% of the total variance, showed strong negative correlations with several *Lactobacillus* species (*L. alvi*, *L. johnsonii*, *L. delbrueckii,* and *L. salivarius*), as well as with *Clostridium disporicum* and *Romboutsia timonensis*. Component 2, accounting for 25.68% of the variance, was strongly positively correlated with *L. kitasatonis, C. jejuni,* and *L. pontis*. Component 3, responsible for 16.49% of the variance, was most strongly correlated with *Aeriscardovia aeriphila* and *Romboutsia* sp., indicating that these species are the main contributors to this component.

The graph shown in Figure 4 presents the distribution of samples along the first two principal components (PC1 and PC2), which together explain 80.46% of the total variance in the data. Component 1 (horizontal axis) separated samples associated with *Lactobacillus species*, *Clostridium disporicum,* and *Romboutsia timonensis*, which exhibited strong negative correlations. Component 2 (vertical axis) differentiated treatments with higher abundances of *Lactobacillus kitasatonis*, *Campylobacter jejuni*, and *Lactobacillus pontis*, species that were strongly positively correlated.

Treatments identified by abbreviations: SS500 = single strain with 500 g/t included in basal ration; SS1000 = single strain with 1000 g/t included in basal ration; B200 = blend in dosage (200 g/t) included in basal ration; and B400 = blend in dosage of 400 g/t included in basal ration; source: prepared by the authors.

The analysis allowed visualization of clustering according to treatments, indicating that changes in microbial composition were captured by the principal axes. Treatments with a higher predominance of *Lactobacillus* tended to cluster in the negative region of PC1, while those influenced by *Campylobacter* and *L. kitasatonis* were in the positive region of PC2. This separation highlights the contribution of dietary treatments to structuring the intestinal microbial community.

## 4. Discussion

The intestinal microbiota of laying hens can be influenced by animal density, species diversity, and factors related to the rearing system, daily management, diet composition, species, housing conditions, litter hygiene, population density, and environmental conditions (air temperature and humidity) [23]. The intestinal dysbiosis, defined as an imbalance between beneficial and pathogenic microorganisms in the gastrointestinal tract, can compromise digestion, nutrient absorption, and mucosal integrity, ultimately reducing the productive performance of laying hens [4,24].

In this experiment, the relationship between probiotic addition and intestinal microbial balance was clearly demonstrated. The inclusion of different strains and dosages of *Bacillus subtilis* and a probiotic blend containing multiple *Lactobacillus* and *Enterococcus* species aimed to modulate the small intestinal microbiota and prevent dysbiosis [25]. The results showed that the diet with the probiotic blend (200 g/t) promoted the growth of beneficial bacteria such as *Lactobacillus crispatus* and *L*. *aviarius*, indicating a healthier and more stable microbial community. Conversely, the exclusive inclusion of a single strain of *Bacillus subtilis* was associated with an increase in *Helicobacter brantae*, which may reflect an imbalance in the intestinal ecosystem.

Similar trends have been reported in previous studies where probiotic blends were more effective than single strains in promoting eubiosis and intestinal health. Therefore, the present findings reinforce the importance of using multi-strain probiotics as a strategy to maintain intestinal eubiosis and improve the gastrointestinal health and productivity of laying hens [23,26].

The presence of pathogenic microorganisms in the hen’s intestine can interfere with digestion, nutrient absorption, and overall physiological function, negatively affecting enzymatic activity, epithelial integrity, and intestinal function; therefore, the microbiota impacts productive performance and mortality in laying hens [26].

For decades, antibiotics were used as growth promoters in the poultry industry to modulate the intestinal microbiota. With the reduced use of antibiotics in poultry production, there is growing interest in identifying probiotic organisms and prebiotic preparations that can protect laying hens and enhance productivity [27]. Probiotics can increase the production of volatile fatty acids (VFAs), mainly acetate, propionate, and butyrate, which are directly absorbed in the intestine and serve as an energy source for tissues [28].

Probiotics are defined as “live microbial dietary supplements that beneficially affect the host by improving its intestinal microbial balance” and are composed of live microorganisms, including bacteria, fungi, and yeasts [29,30]. The microorganisms used as probiotics are generally lactic acid bacteria, non-lactic acid bacteria, and yeasts, with the most common genera being Lactobacillus, Bifidobacterium, Escherichia, Enterococcus, and Bacillus [31].

The phylum Proteobacteria predominates in the early life stages of laying hens and is associated with activation of the intestinal immune response and promotion of immune system development [32,33]. Members of the genus Bacillus exhibit diverse physiological characteristics, including mesophilic, facultative or obligate thermophilic, extreme thermophilic, psychrophilic, and halophilic capabilities, enabling them to grow under extreme temperature, pH, and salinity conditions [34].

In commercial poultry, prevalent genera include *Enterococcus*, *Lactobacillus*, *Escherichia*, *Shigella*, *Pseudomonas*, other members of the Enterobacteriaceae family, Mycoplasma, Staphylococcus, and Streptococcus [35]. Among these, *Enterococcus* and *Lactobacillus* are resident probiotic bacteria capable of adhering to host cells, excluding or reducing pathogenic bacterial adherence, persisting, and multiplying; they also produce acids, hydrogen peroxide, and bacteriocins, preventing pathogenic bacterial growth and maintaining microbiota balance [36,37].

In the present study, the classified phyla showed a predominance of Bacillota in treatment with basal ration, while the addition of probiotics led to a predominance of Proteobacteria in single-strain treatments and Bacillota in blended probiotic treatments. The increase in Bacillota is associated with antibiotic-like effects and includes beneficial genera such as Clostridium and Lactobacillus, which contribute to immunomodulation through fermentative activity on caloric feeds like corn [38,39,40,41].

Similar results were reported by Ferreira et al. (2024) [42], who quantified Proteobacteria at 3.55% and Actinobacteriota at 2.31% in the intestinal microbiota of laying hens treated with antibiotics and natural performance enhancers under heat stress. Chen et al. [43] found that commercial Roman layers exhibited significantly lower diversity than native chickens, with predominance of the phyla Firmicutes, Bacteroidota, and Proteobacteria. The genus Romboutsia was most abundant, and Fusobacterium was the dominant strain in Luning chickens (93.36%).

The genus Lactobacillus (family Lactobacillaceae) efficiently ferments carbohydrates, lowering intestinal pH and impacting the growth of other bacterial species [44]. Probiotics composed of *Lactobacillus* and *Bifidobacterium* are widely used because they modify the microbiota, enhance short-chain fatty acid (SCFA) production, strengthen the immune system, and provide enzymes not produced by the host [45,46]. Increased SCFA stimulates mucin secretion by epithelial cells and increases local permeability [47].

VFAs produced by bacterial fermentation promote epithelial cell proliferation, directly regulate gene expression and phenotype, and act as epigenetic regulators of multiple genes [48,49,50]. In this context, probiotic bacteria such as *Lactobacillus* spp. and *Bacillus* spp. stimulate VFA production in poultry [51,52]. Butyrate regulates nutrient allocation from the liver and adipose tissue to muscle, acting as a muscle insulin receptor (IRβ) [53].

The diversity of the microbiota in different sections of the intestine is due to changes in the composition of nutrients provided by the ingredients available during enzymatic digestion and absorption through the intestinal wall [53]. Thus, in the small intestine, the duodenal portion shows a higher proportion of Firmicutes, followed by Bacteroidetes, with Proteobacteria being the predominant phylum [54].

This occurs because the duodenum has a low transit density and low pH. In this region of the small intestine, the dilution of pancreatic and biliary secretions occurs slowly, leading to a considerable delay in microbial stabilization (approximately two weeks). The microbiota is mainly composed of *Lactobacillus* sp., *Enterococcus* sp., *and Escherichia coli* [55].

In the distal portion of the small intestine, the ileum exhibits a differentiated microbial development with the presence of the genera Lactobacillus, Streptococcus, Enterobacteriaceae, and Clostridiaceae [56]. The dominant genus in the ileum is Enterococcus, accounting for about 30% [57]. Firmicutes represent approximately 85% of the microbiota, followed by Proteobacteria. There are also dominant orders such as Clostridiales, Bacilli, Lactobacilli, and Gammaproteobacteria, mainly Enterobacteriales [58].

The small intestine is primarily colonized (95%) by lactic acid-producing bacteria, mainly *Lactobacillus* spp., *Enterococcus* spp., and *Streptococcus* spp., which have an exclusively fermentative metabolism [59,60,61]. In the studied diets, probiotic inclusion affected *Helicobacter* and *Lactobacillus* species. Peng et al. [62] reported that carbohydrate-fermenting butyrate-producing bacteria enhance intestinal barrier function and are highly related to feed efficiency in poultry; thus, carbohydrate type influences the Bacillota/Bacteroidota ratio and gut microbial diversity [63].

Fiber-fermenting bacteria in the colon produce SCFAs (acetate, propionate, and butyrate), regulating metabolic processes, intestinal motility, nutrient absorption, mucosal function, and energy production via lipogenesis, gluconeogenesis, and cholesterol synthesis [64,65,66,67]. Laying hens undergo significant physiological changes during the laying period, increasing lipid metabolism and nutrient utilization, which drives shifts in gut microbiota [68].

Bacteria of the genus Acinetobacter vary in function, with some species common under pathogenic conditions. *Actinobacteria* are Gram-positive, non-spore-forming, non-motile, strictly anaerobic bacteria [69]. In broilers, genera such as Helicobacter, Campylobacter, and Clostridium are common; some *Clostridium* species are pathogenic (*C. perfringens*, *C. tetani*), but most are commensals contributing to host physiology and immunity [70].

The genus Turicibacter is associated in the intestine with bile acid metabolism and, consequently, with lipid metabolism, in addition to being involved in intestinal immune modulation [71]. Its presence and potential migration into the reproductive tract have recently been investigated, and increased abundance has been associated with lower productivity, advanced age in laying hens, and reduced egg quality [72]. Currently, Turicibacter is considered a biomarker of reproductive health, and its abundance can be monitored through microbiota profiling [73]. In the present experiment, a dose–response modulation of Turicibacter abundance was observed in response to probiotic inclusion, with the lowest relative abundances detected in treatments with lower inclusion levels (500 g/t single-strain and 200 g/t blend).

The results indicated that single-strain and multi-strain probiotics exerted distinct effects on the intestinal microbiota of laying hens. The predominance of *Helicobacter brantae* in treatments with *Bacillus subtilis* alone suggests a possible dysbiosis or selective microbial imbalance [74]. In contrast, the probiotic blend promoted greater diversity and abundance of beneficial bacteria, demonstrating synergistic effects among strains [27]. Therefore, multi-strain formulations proved to be more effective in maintaining eubiosis and promoting intestinal health in laying hens.

## 5. Conclusions

The present study demonstrates that probiotic addition in diet can modulate the small intestinal microbiota of Isa Brown laying hens raised in a cage-free system, although the magnitude and direction of these effects depend on the probiotic formulation and dosage. High inclusion levels of the single Bacillus subtilis strain markedly increased the abundance of *Helicobacter brantae*, suggesting that this strain alone may shift the microbial community toward a less desirable profile under the conditions evaluated. In contrast, the probiotic blend at 200 g/t promoted a more balanced and potentially beneficial microbial structure, characterized by higher proportions of *Lactobacillus crispatus* and *Lactobacillus aviarius*. These findings indicate that multi-strain probiotic formulations, particularly at moderate inclusion levels, may better support intestinal microbial stability in cage-free laying hens. Further research is warranted to elucidate the functional implications of these microbial shifts and their long-term effects on hen health and productivity.

## Figures and Tables

**Figure 1 animals-15-03388-f001:**
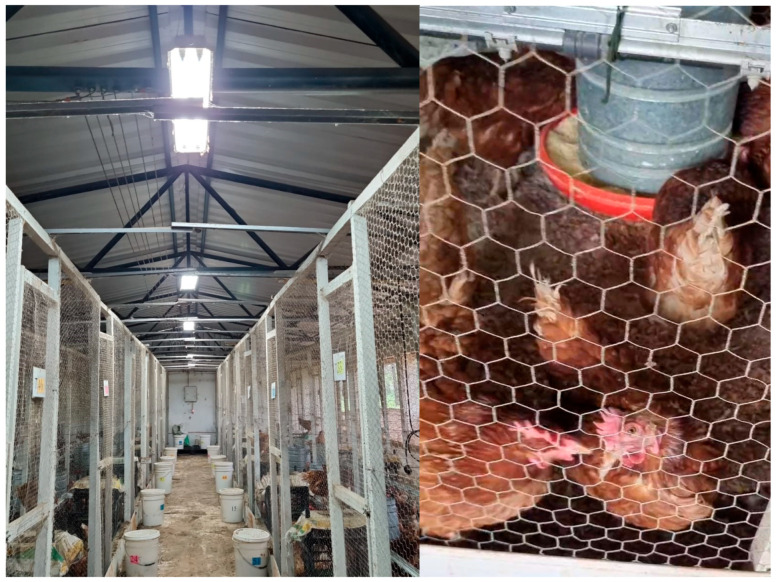
Laying hen facilities source: personal archive.

**Figure 2 animals-15-03388-f002:**
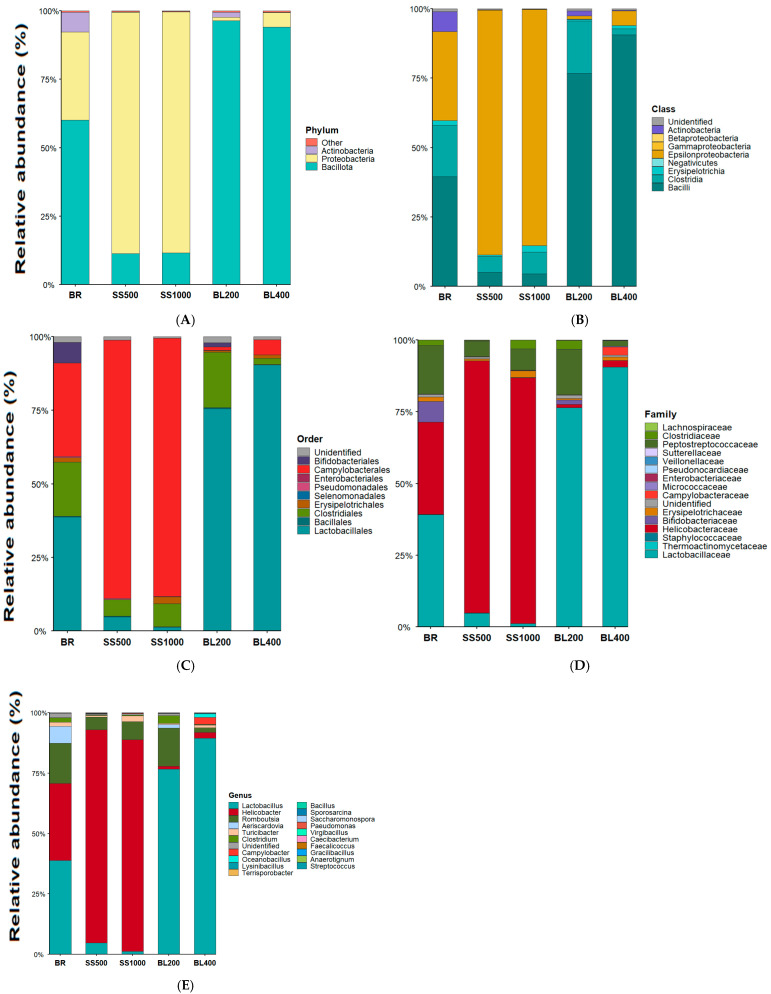
Treatment-wise relative abundance of the gut microbiota (phylum to genus). Panels (**A**–**E**) show stacked relative abundances (means across replicates) at phylum, class, order, family, and genus levels, respectively. Treatments: BR = basal ration; SS500/SS1000 = single-strain probiotic at 500/1000 g·t^−1^; BL200/BL400 = probiotic blend at 200/400 g·t^−1^. source: prepared by the authors.

**Figure 3 animals-15-03388-f003:**
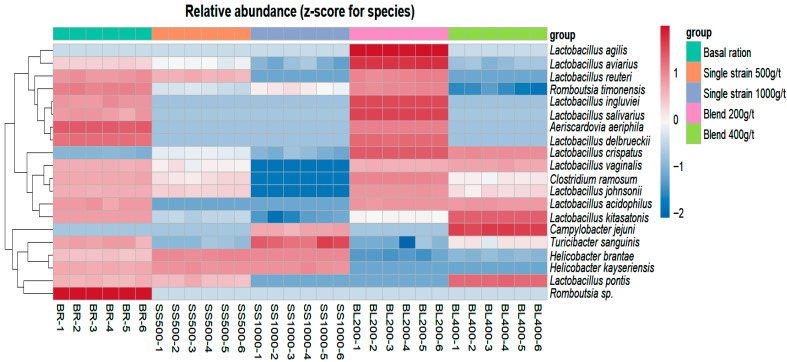
Relative abundance of the ratio of microorganism species found in treatments. Treatments according to repetitions identified by abbreviations: BR = basal ration; SS = single strain; BL = blend and their respective dosages (200 g/t; 400 g/t; 500 g/t; 1000 g/t), the numbers indicate the number of replicates for each treatment. Source: prepared by the authors.

**Figure 4 animals-15-03388-f004:**
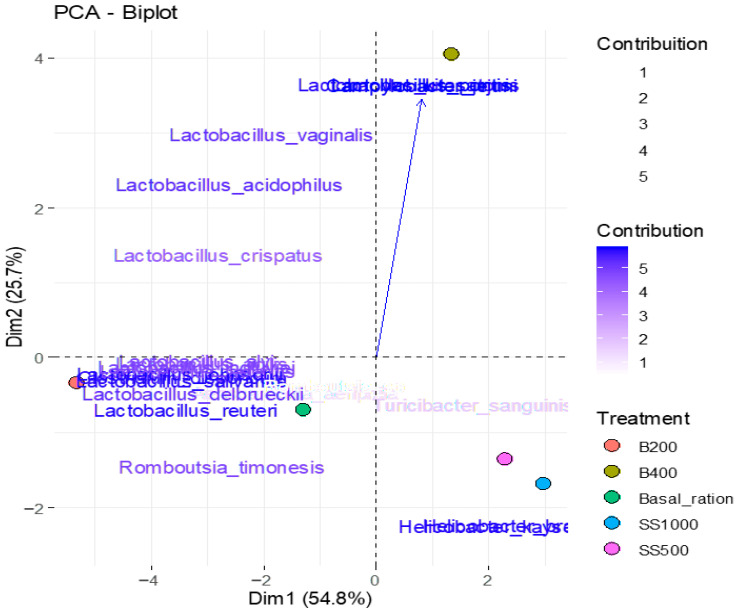
Biplot of the principal component analysis (PCA) of microbial variables.

**Table 1 animals-15-03388-t001:** Proximate composition and calculated nutritional levels of the diets.

Ingredients	Basal Ration	Single Strain ^1^ (500 g/t)	Single Strain ^1^ (1000 g/t)	Blend ^2^ (200 g/t)	Blend ^2^ (400 g/t)
Corn 7.88%	66.021	66.021	66.021	66.021	66.021
Soybean meal 45%	20.151	20.151	20.151	20.151	20.151
soybean oil	1.708	1.708	1.708	1.708	1.708
Limestone	9.992	9.992	9.992	9.992	9.992
Dicalcium phosphate	0.730	0.730	0.730	0.730	0.730
Mineral premix	0.500	0.500	0.500	0.500	0.500
Common salt	0.388	0.388	0.388	0.388	0.388
L-lysine	0.138	0.138	0.138	0.138	0.138
DL-methionine	0.274	0.274	0.274	0.274	0.274
L-threonine	0.084	0.084	0.084	0.084	0.084
L-tryptophan	0.018	0.018	0.018	0.018	0.018
Total (kg)	100.0	100.0	100.0	100.0	100.0
Calculated value					
ME (Mcal/kg)	2.850	2.850	2.850	2.850	2.850
Crude protein (%)	14.830	14.830	14.830	14.830	14.830
Linoleic acid (%)	1.514	1.514	1.514	1.514	1.514
Total Lysine (%)	0.747	0.747	0.747	0.747	0.747
Total methionine (%)	0.688	0.688	0.688	0.688	0.688
Total threonine (%)	0.576	0.576	0.576	0.576	0.576
Total tryptophan (%)	0.172	0.172	0.172	0.172	0.172
Calcium (%)	4.100	4.100	4.100	4.100	4.100
Potassium (%)	0.568	0.568	0.568	0.568	0.568
Phosphorus (%)	0.213	0.213	0.213	0.213	0.213
Sodium (HCL) (%)	0.175	0.175	0.175	0.175	0.175

Mineral premix, %/kg of feed: Methionine (min.) 170 g/kg; Iron (min.) 6700 mg/kg; Copper (min.) 1.440 mg/kg; Manganese (min.) 144 g/kg; Zinc (min.) 12 g/kg; Iodine (min.) 288 mg/kg; Selenium (min.) 48 mg/kg; Vitamin A (min.) 1.600 IU/kg; Vitamin D_3_ (min.) 500.000 IU/kg; Vitamin E (min.) 2.400 IU/kg; Vitamin K_3_ (min.) 440 mg/kg; Vitamin B_1_ (min.) 200 mg/kg; Vitamin B_2_ (min.) 800 mg/kg; Vitamin B_6_ (min.) 300 mg/kg; Vitamin B_12_ (min.) 2.440 µg/kg; Niacin (min.) 6.000 mg/kg; Calcium pantothenate (min.) 2.000 mg/kg; Folic acid (min.) 100 mg/kg; Biotin (min.) 10 mg/kg; Choline chloride (min.) 42 g/kg; Zinc bacitracin (min.) 2.200 mg/kg. ^1^ feed with a single strain consisting of: *Bacillus subtilis* (1.0 × 10^8^ CFU/g). ^2^ feed containing strains of: *Bacillus subtilis* (1.0 × 10^8^ CFU/g); *Enterococcus faecium* (6.0 × 10^8^ CFU/g); *Lactobacillus acidophilus* (1.0 × 10^8^ CFU/g); *Lactobacillus delbrueckii* (1.0 × 10^8^ CFU/g); *Lactobacillus plantarum* (3.0 × 10^8^ CFU/g); *Lactobacillus reuteri* (5.0 × 10^8^ CFU/g); *Lactobacillus salivarius* (1.0 × 10^8^ CFU/g); *Pediococcus acidilactici* (3.0 × 10^8^ CFU/g).

**Table 2 animals-15-03388-t002:** Means of the proportions of bacteria found in the digesta of the laying hens’ small intestine.

Species	Basal Ration	Single Strain (500 g/t)	Single Strain (1000 g/t)	Blend (200 g/t)	Blend (400 g/t)	*p* Value	EMP ^1^	CV ^2^ (%)
*Aeriscardovia aeriphila*	7.140a	0.000c	0.000c	1.445b	0.000c	0.000110	0.052	7.54
*Campylobacter jejuni*	0.000b	0.000b	0.053b	0.000b	2.795a	0.000091	0.083	5.82
*Clostridium disporicum*	0.860b	0.143c	0.000d	2.531a	0.083c	0.000029	0.026	9.05
*Helicobacter brantae*	31.60b	86.038a	85.500a	1.163c	2.423c	0.000010	0.641	3.80
*Helicobacter kayseriensis*	0.801b	2.41a	2.47a	0.000c	0.000c	0.000012	0.019	11.47
*Lactobacillus acidophilus*	0.590a	0.000b	0.000b	0.660a	0.626a	0.000015	0.332	11.67
*Lactobacillus agilis*	0.000b	0.000b	0.000b	0.730a	0.000b	0.000020	0.009	15.62
*Lactobacillus alvi*	1.050a	0.225b	0.041c	1.051a	0.280b	0.000280	0.021	9.89
*Lactobacillus aviarius*	1.031c	1.416b	0.281d	20.878a	0.123d	0.000222	0.115	5,94
*Lactobacillus crispatus*	0.352d	1.668c	0.358d	36.466a	15.595b	0.000214	0.186	4.18
*Lactobacillus delbrueckii*	1.405b	0.000c	0.000c	1.583a	0.000c	0.000690	0.029	12.23
*Lactobacillus ingluviei*	0.067b	0.000c	0.000c	1.028a	0.000c	0.000901	0.005	6.16
*Lactobacillus johnsonii*	3.125b	0.811c	0.000d	7.803a	0.558c	0.000009	0.104	10.44
*Lactobacillus kitasatonis*	14.15b	0.393d	0.043e	1.656c	67.585a	0.000024	0.100	1.47
*Lactobacillus pontis*	0.181b	0.146b	0.000c	0.000c	3.571a	0.000018	0.025	7.85
*Lactobacillus reuteri*	0.454b	0.143c	0.000d	0.763a	0.000d	0.000015	0.029	6.73
*Lactobacillus salivarius*	0.101b	0.000c	0.000c	3.091a	0.000c	0.000014	0.110	4.27
*Lactobacillus vaginalis*	0.910b	0.106c	0.000c	0.841b	1.218a	0.000011	0.038	5.42
*Romboutsia* sp.	0.935a	0.000b	0.000b	0.000b	0.000b	0.000010	0.111	4.96
*Romboutsia timonesis*	16.85a	5.271d	7.445c	15.878b	1.923e	0.000010	0.174	4.51
*Turicibacter sanguinis*	1.625b	0.625d	2.393a	0.531d	1.108c	0.000024	0.056	11.01

Means followed by the same letter in a row differ significantly according to Fisher’s least significant difference (LSD) test at 5% probability. ^1^ standard error; ^2^ coefficient of variation. Source: prepared by the authors.

**Table 3 animals-15-03388-t003:** Correlation of microbial variables with the first three principal components, eigenvalues, explained variance, and cumulative variance.

Variables	Component 1	Component 2	Component 3
*Lactobacillus_alvi*	−0.91	−0.07	0.40
*Clostridium_disporicum*	−0.98	−0.13	−0.16
*Lactobacillus_crispatus*	−0.79	0.33	−0.50
*Helicobacter_brantae*	0.71	−0.69	−0.10
*Lactobacillus_aviarius*	−0.89	−0.11	−0.43
*Lactobacillus_johnsonii*	−0.99	−0.12	−0.12
*Lactobacillus_kitasatonis*	0.16	0.97	0.15
*Romboutsia_timonesis*	−0.78	−0.47	0.41
*Lactobacillus_vaginalis*	−0.52	0.78	0.34
*Campylobacter_jejuni*	0.23	0.97	−0.04
*Lactobacillus_pontis*	0.23	0.97	0.00
*Lactobacillus_acidophilus*	−0.74	0.60	0.31
*Turicibacter_sanguinis*	0.49	−0.23	0.42
*Aeriscardovia_aeriphila*	−0.40	−0.19	0.89
*Lactobacillus_agilis*	−0.88	−0.08	−0.46
*Lactobacillus_ingluviei*	−0.90	−0.09	−0.40
*Lactobacillus_delbrueckii*	−0.92	−0.20	0.34
*Lactobacillus_salivarius*	−0.98	−0.14	−0.12
*Lactobacillus_reuteri*	−0.96	−0.25	0.05
*Helicobacter_kayseriensis*	0.71	−0.69	−0.13
*Romboutsia_sp*	−0.21	−0.17	0.96
Eigenvalue	11.50	5.39	3.46
% Variance	54.78	25.68	16.49
% Cumulative Variance	54.78	80.46	96.95

Source: prepared by the authors.

## Data Availability

The original contributions presented in this study are included in the article. Further inquiries can be directed to the corresponding author.

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
