# Peer review of "Intestinal Microbiota of Commercial Laying Hens in a Cage-Free System Fed with Probiotics"

_animals, 2025, doi:10.3390/ani15233388_

Round 1

Reviewer 1 Report

Comments and Suggestions for Authors

This study investigated microbiota modulation under cage-free systems, which is a timely topic. However, the author did not address its novelty or the significance in the introduction.

The paper only used descriptive percentages without sufficient alpha/beta diversity indices or statistical comparison of community structures. It appears that the single-strain and blend treatments had opposite effects on the microbiota, which was not discussed. The high dominance of helicobacter brantae in single-strain treatments suggests potential dysbiosis and the results should be critically interpreted. Overall, the discussion is too general and superficial and is not well-linked to the results.

This is a 16S microbiome analysis. Why weren’t cecal samples tested, since bacteria are mostly abundant in the ceca? Although different locations of the intestine were sampled, the results did not reflect the locations. 

The relative humidity seems too high for the birds. However, the author stated that the environmental conditions were within acceptable limits.

The manuscript contains many formatting issues, such as double spacing, double commas, and unnecessary spaces between words. The label in Figure 1 is not in English. There are many mistakes that could have been avoided, suggesting that the manuscript was not prepared to the level for submission. Here are a few examples. 

Line 144: using and employing are repetitive.

Line 178: It should be 7.25% and 1.83% not 7,25% and 1,83%.

Line 180: basal diet treatment, not basal diet. The same applies to the rest of the context.

Author Response

We sincerely thank the reviewer for the time and effort dedicated to evaluating our manuscript. The insightful comments and constructive suggestions greatly contributed to improving the clarity, quality, and scientific rigor of our work. 

Attached is the file presenting the changes made, with the corresponding line numbers where the corrections occurred. The modifications in the article are highlighted in red.

Reviewer 2 Report

Comments and Suggestions for Authors

REVIEWER COMMENTS:

The study ‘Intestinal microbiota of commercial laying hens in a cage-free system fed with probiotics’ conducted by Dr. da Silva and its groups, and reported a variety of interesting results and providing valuable information to relevant poultry research fields. I have read the paper wholeheartedly, and here are some comments and questions:

Minor comments and questions:

[Summary/Abstract]

L44: change the term ‘bird’ to be a scientific term ‘laying hen’ or ‘Isa Brown’ or even ‘layer’ throughout the manuscript.

L45-46 :which strain of 1000g/t?

L46 : which blend of 400 g/t?

L47: define the exact location of the small intestine, it is too vague.

L48-49: Increased compared to which groups?

L49: what is the 86%? and 87%? Just simply based on the abstract, it is a completely unable to understand these levels.

L54: check the typo and add two more keywords

[Introduction]

L60: delete period

L63: space between word and reference

L66: space between word and reference

L67: delete space

L70: delete comma

L70: delete space

L74: spathogenic?

L75: No italic to typhimurium and T should be capitalized.

L76: delete space

L77: delete comma

L86: delete space

> I will not check for these minor errors, sentence rules, or sentence typos after this section.

> Before submitting a paper, this is a basic point that a researcher must check.  

[Materials and Methods]

L93: What is the meaning of the ‘approximately 19 weeks old’? Are you saying that not all layers are 19 weeks old?

Major comments/questions 1: [Materials and Methods]

L93-98: The current experiment is testing the effectiveness of diet in cage-free rearing environments rather than the cage study commonly utilized in laying hens. Therefore, a very specific description of the relatively specific cage-free environment relative to the cage environment is required, which must be described by subheadings in Materials and Methods.

L93-94: ‘with flock uniformity ensured through body weight standardization’. How? Explain here and express standard deviation of BW of layers.

L95-97: Make sure to mark the full rearing area of the pen. It should include the height, too.

Be sure to specify how many drinkers and feeders and nest they have.  

Indicate the size of the drinker, feeder, and nest.

L99-100: How did the natural and artificial light combine here? Mark specifically.

L104-106: How do the readers know that the rearing environment is acceptable with this information? Please indicate the reference.

What is the minimum temperature?

L110: corn-soybean meal-based basal diet

L111-115: indicate the company name and detailed address to CCBC 1001, 1003, and 1004.

[Table 1]

> In total (kg), make 100.0 to be centered of the table.

> Make an additional line: calculated value between the Total (kg) and ME.

[Materials and Methods]

L141: indicate primer information

[Results]

L168-170: Exactly which part of the small intestine? Jejunum? Ileum?

[Figure 1]

> Following from the above, assuming this is the microbiome in the intestinal lumen as you mentioned, how can the relative abundance of Bacillota appear as 0%? Is it an analytical error? It must be explained.

> Which bacteria group is the largest portion in the blend 400 group? The percentage of Proteobacteria shown in the figure is only 5.3%, and the other group was not detected at all.

Major comments/questions 2: [Figure 1, 2, 3, 4, and 5]

> I don't understand why these pictures are all presented in different formats.

Except for the species in Figure 6, please unify the expression methods in Figure 1-5 so that readers can easily understand the differences between groups.

[Figure 4 and 5]

> Need a correction in treatment name, basal diet or basal ration?  

[Table 2]

> Need translation.

> How the p-value can be = 0.000 in several species? For example, Helicobacter brantae Lactobacillus alvi Lactobacillus aviarius Lactobacillus crispatus Lactobacillus kitasatonis, etc.

> Abbreviation of EMP? And what is the unit?

> CV, %?

[Figure 6]

> Explain the abbreviation such as BR-1, and SS500, to the figure legend.

[Figure 7]

> Need a correction in treatment name, and add abbreviation in the legend.

Major comments/questions 3: [All Tables and Figures]

> Tables and figures have no legends. There should be minimal description added to the figures and tables.

[Discussion]

L373-375: What does intestinal dysbiosis have to do with this experiment? Are there any infections or challenges that could cause an imbalance in the gut health and gut microbiota of laying hens?

L375-378: References?

L378-378: If you mentioned that age affects the microbiota of chickens, shouldn't you give an example of a specific age, for example, 65 weeks or more?

L386-388: References?

L427-429: Does this sentence correspond to intestinal health promotion? Please check again.

L453-455: References?

L457-459: References?

Major comments/questions 4: [Discussion]

> Various beneficial effects from the addition of probiotics, such as SCFA enhancement, immune response enhancement, increased nutrient absorption ability, increased energy production, higher protein synthesis, etc., have not been analyzed at all in the present experiment. Consequently, your discussion section should be written simply explaining the microbiota changes in intestinal health rather than comparing the benefits of previous studies. A substantial amount of discussion section content will need to be deleted/modified.

Major comments/questions 5: [Conclusion]

L494-496: This experiments discuss in depth the relationship between cage-free layers and probiotic addition in laying hens. However, no performance-related parameters are not presented here. This indicates that intestinal health and performance cannot be mentioned in the conclusion section. Therefore, conclusion must be changed ‘The 200 g/t probiotic blend inclusion showed the most favorable modulation of the small intestinal microbiota and enhancing the microbial balance’ If you disagree, please provide your opinion.

Author Response

(The authors gave the same response as above.)

Round 2

Reviewer 1 Report

Comments and Suggestions for Authors

No more comments. 

Author Response

Thank you very much for your careful review and valuable comments on our manuscript. We greatly appreciate the time and effort you spent to provide suggestions.

Reviewer 2 Report

Comments and Suggestions for Authors

Significant efforts appear to have been added to the revision, and the modifications have been well reflected.

Author Response

(The authors gave the same response as above.)
